# O-Band Frequency-Tunable (10–22 GHz) Ultra-Low Timing-Jitter (<12-fs) Regenerative Mode-Locked Laser

Hefei Qi [1,2,3], Zhihao Zhang [1,2,3], Dan Lu [1,2,3,*], Ruikang Zhang [1,2,3] and Lingjuan Zhao [1,2,3]

1   Key Laboratory of Semiconductor Materials Science, Institute of Semiconductors, Chinese Academy of Sciences, Beijing 100083, China; qihefei@semi.ac.cn (H.Q.); zhangzhihao20@semi.ac.cn (Z.Z.); rkzhang@semi.ac.cn (R.Z.); ljzhao@semi.ac.cn (L.Z.)
2   Center of Materials Science and Optoelectronics Engineering, University of Chinese Academy of Sciences, Beijing 100049, China
3   Beijing Key Laboratory of Low Dimensional Semiconductor Materials and Devices, Beijing 100083, China
*   Correspondence: ludan@semi.ac.cn

**Abstract:** A frequency-tunable and low-timing-jitter O-band regenerative mode-locked laser (RMLL) using an optoelectronic oscillation configuration and electric-controlled yttrium iron garnet (YIG) bandpass filter is proposed and demonstrated. In this scheme, an O-band semiconductor optical amplifier (SOA) is used as the gain medium of the RMLL to realize a dispersion-management-free operation during frequency tuning. With a polarization-maintaining fiber loop of 300 m, we produced a robust frequency-tunable RMLL with a pulse width below 16 ps, phase noises below −123 dBc/Hz at a 10-kHz frequency offset from the carrier frequency, and timing jitter less than 12 fs (integrated in 1-kHz to 1-MHz range) in a frequency tuning range between 10 GHz and 22 GHz.

**Keywords:** mode-locked lasers; regenerative mode locking; optical short pulse; optoelectronic oscillator; microwave photonics

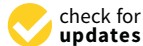



## 1. Introduction

High-frequency and low-timing-jitter optical pulse sources are highly desirable in the fields of high-speed photonic analog-to-digital converters (ADC), optical communication systems and optical information processing systems [1–3]. As a typical short optical source, the fiber-based mode locked laser (MLL) has been extensively studied due to its ability to generate short pulses with an easily accessible setup in the laboratory environment [4,5]. To realize a high-frequency and high-quality optical pulse output, the active mode-locking technique using a low-noise external RF source is usually adopted. The phase noise and the timing jitter of the mode-locked laser are determined by the phase noise of the RF source. Furthermore, to obtain a frequency-tunable low-timing-jitter optical pulse output, high-quality synthesizers need to be used to adjust the repetition rate, resulting in bulky and expensive systems. An alternative way to realize active-mode locking is the regenerative mode-locking technique [6,7], in which the driving RF signal is derived from the amplified mode-beating signal of the MLL. This scheme requires no external RF source and is capable of generating optical pulses with ultra-low phase noise, which are even superior to those of the most advanced RF synthesizers, in which an optoelectronic oscillation fiber loop is adopted [8,9]. This high performance can be attributed to the enhancement of the effective *Q* factor of the oscillation cavity due to the combined contribution of the long cavity length and appropriate compensation of oscillation loss through the optical gain. A regenerative MLL (RMLL) with a long cavity configuration is also applied as a coupled optoelectronic oscillator. RMLL provides a cost-effective solution for generating high-repetition-rate short optical pulses with low phase noise and timing jitter.

Currently, most RMLLs work in the 1.5-micrometer-band, using an erbium-doped fiber amplifier (EDFA) or a semiconductor optical amplifier (SOA) to provide optical

gain. During pulse formation and transmission through the laser cavity, fiber dispersion must be taken into consideration. A dispersion management scheme using a dedicated combination of single-mode fiber (SMF) and dispersion compensation fiber (DCF) or dispersion-shifted fiber (DSF) should be adopted to assist soliton formation [7,10] as well as optical spectrum broadening [11]. Furthermore, the repetition rates of RMLLs are mostly fixed and determined by the center frequency of the electrical bandpass filter (EBF). With the development of the short-reach optical access network and silicon photonics, the O-band (1.3 μm) channel is becoming increasingly important. Optical sampling and information processing in the O-band will be an important area of research. Compared to the 1.5-micrometer band, O-band optical pulses experience a lower dispersion when travelling through the optical fiber, which relaxes the dispersion management requirement when constructing an RMLL. However, research on the O-band RMLL is still limited.

In this article, we report ultra-low timing-jitter RMLLs based on an O-band SOA. By using a fixed-frequency EBF and a 910-m single-mode fiber cavity, a 25-gigahertz-mode locked short-pulse train with phase noise of −133 dBc/Hz at 10-kHz carrier frequency offset and timing jitter of 3.6 fs (integrated in 1-kHz to 1-MHz range) was obtained. By adopting an electric controlled Yttrium Iron Garnet (YIG) bandpass filter and polarization-maintaining fiber (PMF) of 300 m, a self-starting mode locked output with a pulse width less than 16 ps, tunable from 10–22 GHz was obtained. The phase noises of the frequency-tunable optical pulses are all below −123 dBc/Hz at a 10-kHz offset from the carrier frequency, with a corresponding timing jitter below 12 fs (integrated in 1-kHz to 1-MHz range).

## 2. Principle and Experiment Setup

The schematic diagram of the regenerative mode-locked laser is shown in Figure 1. The laser cavity consists of an O-band SOA to provide the optical gain, a lithium niobate (LiNbO$_3$) Mach–Zehnder intensity modulator as an artificial saturable absorber to provide loss modulation, a spool of optical fiber to increase the $Q$ factor of the laser cavity, an optical bandpass filter to define the oscillation wavelength and prevent multiwavelength mode locked lasing, and two optical isolators that are placed on both sides of the SOA to guarantee the unidirectional travel of the optical signal. Part of the optical signal is tapped out by an optical coupler and converted into an electrical signal by a photodetector, followed by two stages of RF amplification with a total gain of 46 dB and an EBF. The filtered RF signal is then fed into the modulator to enclose the optoelectronic feedback loop and perform regenerative mode-locking. The optical cavity and the optoelectronic feedback cavity form a dual-loop structure, improving the side-mode suppression ratio (SMSR) due to the mutual phase cancellation of the intermediate modes. The optical output of the RMLL is analyzed by an optical spectrum analyzer (OSA) (Advantest Q8384), and an autocorrelator (APE PulseCheck), and the phase noise and timing jitter are analyzed by a phase noise analyzer (Rohde & Schwarz FSWP50, 1 MHz to 50 GHz).

The oscillation frequency of the RMLL is determined by the center frequency of the EBF and is an integer harmonic of the fundamental frequency determined by the mode spacing of the fiber cavity. The regenerative mode-locking tends to self-stabilized, since disturbances of the oscillation frequency in the optical cavity result in an automatic adjustment of the modulation frequency in the feedback loop. The single-sideband noise power spectral density of the resonator can be written as [9]:

$$L(f) = \left[1 + \left(\frac{\nu_0}{2Qf}\right)^2\right] \frac{S'_\varphi(f)}{2} \tag{1}$$

where $\nu_0$ is the oscillation frequency, $\nu_0/2Q$ is the Leeson frequency, and $S'_\varphi(f)$ is the total noise spectral density. A high $Q$ factor will result in the low phase noise performance of the oscillator, and the cavity $Q$ can be expressed as:

$$Q = \omega_0 \tau_R = 2\pi \nu_0 \frac{L'}{\delta c} \tag{2}$$

where $\tau_R$ is the time constant of the cavity, representing the average photon lifetime in the cavity, $L'$ is the optical path length of the cavity, $\delta$ is round-trip loss, and $c$ is the speed of light in vacuum. Adopting a long fiber cavity and compensating the cavity loss using an amplifier results in the improved phase noise and timing jitter performance of the RMLL.

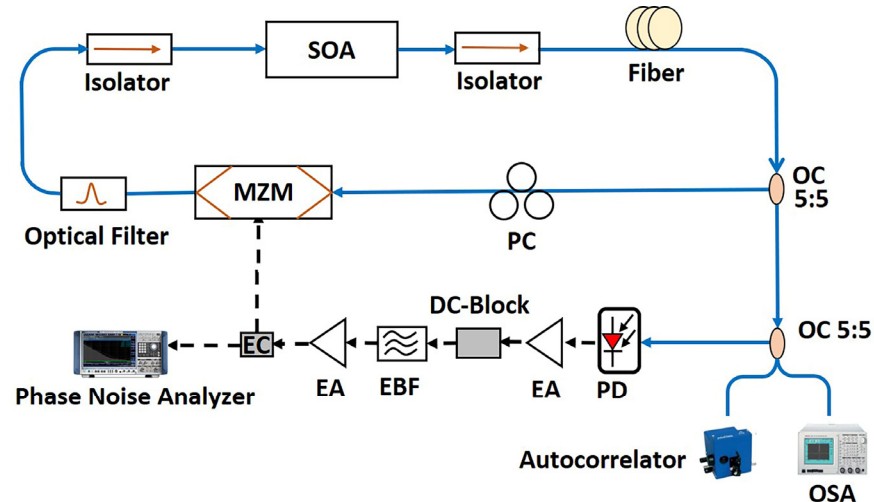

**Figure 1.** Schematic of the regenerative mode-locked laser. SOA: semiconductor optical amplifier; OC: optical coupler; PC: polarization controller; MZM: Mach–Zehnder intensity modulator; PD: photodetector; EA: electrical amplifier; EBF: electrical bandpass filter; EC: electrical coupler; OSA: optical spectrum analyzer.

## 3. Experiment Results

Two experiments were conducted to characterize the performance of the proposed RMLL structure. We first used a single-mode fiber with a fixed frequency EBF to construct a 25-gigahertz RMLL. Next, a tunable YIG filter and polarization maintain (PM) fiber were used to construct a robust, self-starting, repetition-rate-tunable RMLL.

### 3.1. 25-Gigahertz Regenerative Mode-Locked Laser

An SMF-based RMLL was constructed using different fiber lengths to exploit the pulse and phase noise performance of the O-band output. During the experiment, the center wavelength of the RMLL was around 1337 nm, corresponding to a fiber dispersion of 1 ps/nm. The center frequency and the bandwidth of the EBF were 25 GHz and 20 MHz, respectively. During the experiment, the SOA operated at an output power of around 11 mW. By adjusting the polarization controller, the fiber laser could be easily kicked into the mode locked state. Figures 2 and 3 shows the typical measurement results from a 320-m optical cavity. The mode-locked optical spectrum shown in Figure 2a demonstrates a clear harmonic mode-locked characteristics of the laser. The autocorrelation trace of the optical pulse is shown in Figure 2c, which well fits the Gaussian shape, corresponding to a pulse width of 9.9 ps. Figure 3a shows the typical RF spectrum of the RMLL. A clean single tone signal at 25 GHz can be observed. The SMSR of the RMLL reached a level of 61 dB, as shown in Figure 3b, where the side-mode spacing of 640 kHz corresponds to the fiber length around 320 m. Due to the fast response of the SOA compared to the duration of the optical pulse [12,13], the high-frequency amplitude noise and super-mode competition were well suppressed with a very stable output.

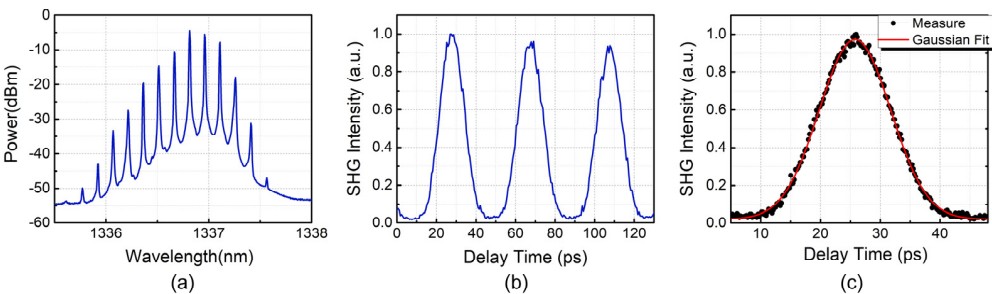

**Figure 2.** The 25-gigahertz regenerative mode−locked laser with 320-m single mode fiber (**a**) optical spectrum; (**b**) autocorrelation trace; (**c**) autocorrelation trace fitting.

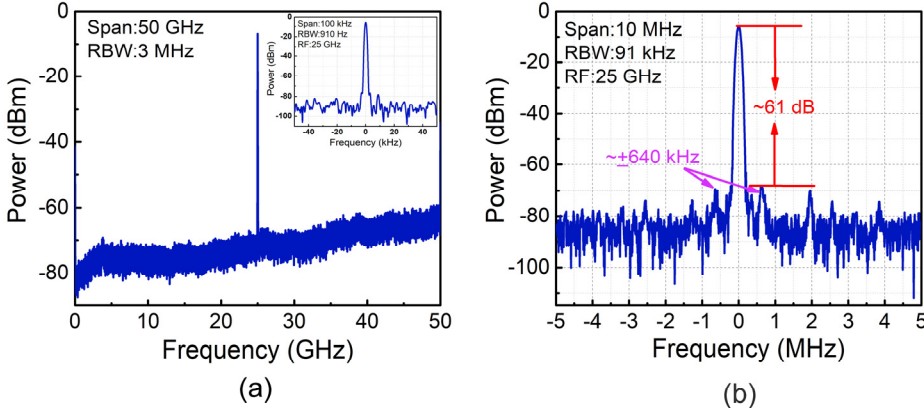

**Figure 3.** The 25-gigahertz regenerative mode-locked laser with 320-m single-mode fiber (**a**) RF power spectrum in a 50-gigahertz and 100-kilohertz (inset) span; (**b**) RF power spectrum in a 10-MHz span.

The phase noise performance of the RMLL with different laser cavity lengths is compared in Figure 4a. At a 10-kilohertz offset from the carrier frequency of 25 GHz, the phase noises for the 45-m, 320-m, and 910-m laser cavities were −104 dBc/Hz, −122 dBc/Hz, and −133 dBc/Hz, respectively. For comparison, the phase noise of the 25-gigahertz signal from an RF signal generator R&S SMA100B is also plotted in the figure. The phase noise performance of the 320-m and 910-m configurations showed a 5-dB and 17-dB improvement, respectively, over SMA100B, starting from a 10-kilohertz offset, which means that the timing jitter performance of the RMLL can be superior to that of an active mode-locked fiber laser driven by the best-in-class signal generator. With the increase in the fiber length, the phase noise at the 10-kilohertz offset decreased at a 20-decibel-per-decade dependence on the fiber length, as shown in Figure 4b. The timing jitter of the pulse also reduced with the increase in the fiber length, and was calculated as 219 fs, 9.51 fs, and 3.6 fs, respectively, for the 45-m, 320-m, and 910-m laser cavities, in an integration range from 1 kHz to 1 MHz. The pulse widths were measured as 10.2 ps (45 m), 9.9 ps (320 m), and 7.3 ps (910 m), all fitting well to a Gaussian pulse shape. A further increase in the fiber length would cause a stability issue with the RMLL, since the fiber-mode spacing became smaller and the gain margin for the main mode over the side mode would not have been significant enough to maintain a stable single-mode oscillation.

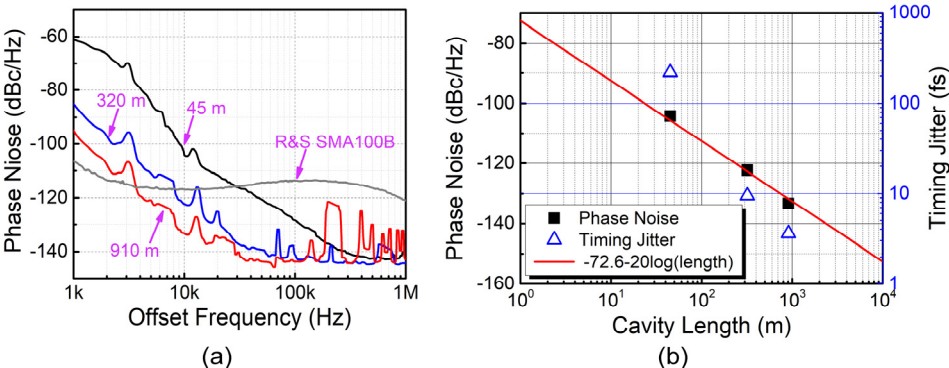

**Figure 4.** The 25-gigahertz regenerative mode−locked laser with 45-m, 320-m, and 910-m single−mode fiber (**a**) phase noises; (**b**) timing jitter (blue triangle), phase noise at a 10-kilohertz offset from the center frequency (black square), phase noise fitting curve of loop length (red line).

### 3.2. Frequency-Tunable Regenerative Mode-Locked Laser

The SMF-based RMLL still required polarization control to facilitate the oscillation. To build a self-starting RMLL with a tunable repetition rate, we adopted a polarization-maintaining structure using PM fiber and PM components, such as isolator, optical couplers, and optical bandpass filter (bandwidth 1 nm). An electric-tuned YIG filter was used to adjust the repetition frequency of the RMLL to form a polarization-maintaining YIG-tuned RMLL (PM-YIG-RMLL). The polarization controller in Figure 1 was removed from this structure. To reduce the timing jitter while maintaining a stable output, the length of the PM fiber was chosen to be 300 m.

During the experiment, the center wavelength of the RMLL was set at about 1333.5 nm. The RMLL started oscillation automatically once the system power was on, without the additional requirement of polarization control. Figure 5 shows the typical RF spectrum and the autocorrelation trace of the RMLL when the YIG filter was tuned to 10 GHz. The RF spectrum shows well suppressed side modes with a SMSR over 70 dB, as shown in Figure 5b. The pulse width was calculated as 12.6 ps, assuming a Gaussian shape, as shown in Figure 5c.

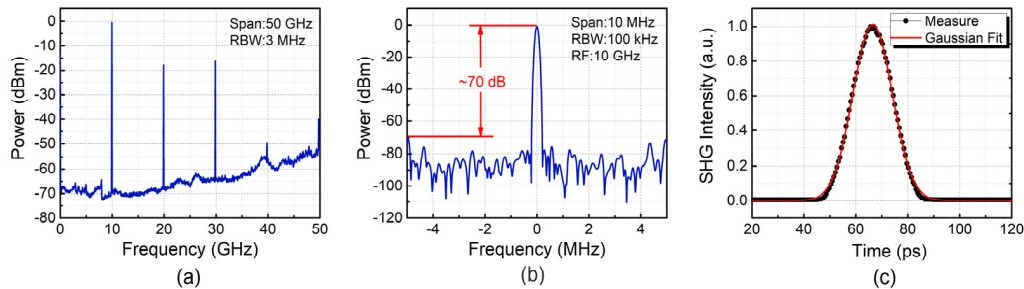

**Figure 5.** The 10-gigahertz microwave signal output of the PM-YIG-RMLL (**a**) RF power spectrum in a 50-gigahertz span; (**b**) RF power spectrum in a 10-MHz span; (**c**) autocorrelation trace fitting.

The YIG filter is a current-tuned device, requiring precise control of the driving current. By adopting a low-noise-current source, the center frequency of the YIG filter can be precisely controlled. By adjusting the driving current of the YIG filter, quick-repetition-rate switching of the mode-locked output can be obtained. Figure 6a shows the measured oscillation frequency of the RMLL vs. the driving current of the YIG filter, which can be linearly tuned from 10–22 GHz with a tuning slope of 75 MHz/mA. Figure 6b shows the mode-locked optical spectrum of the output at several typical repetition frequencies. The overlapped RF spectrum with a tuning step of 1 GHz is shown in Figure 6c.

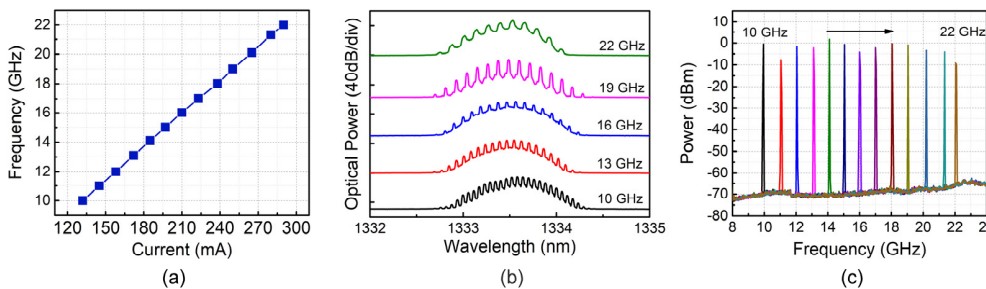

**Figure 6.** (**a**) The relationship between the center frequency of the YIG filter and the driving current; (**b**) overlapped optical spectrum of the PM−YIG−RMLL; (**c**) overlapped RF spectrum of the PM−YIG−RMLL.

Figure 7a shows the phase noise values at the 10-kilohertz frequency offset from the carrier frequency when the RMLL is tuned from 10 GHz to 22 GHz, with an overall phase noise lower than −123 dBc/Hz for all frequencies and the lowest phase noise of −125.7 dBc/Hz at 10 GHz. For a conventional active-mode locked laser driven by a signal generator, its phase noise increases by $20\log(N)$ when the repetition rate is multiplied by $N$. The phase noise of the RMLL's output only demonstrated less than 3-dB degradation when the repetition rate doubled from 10 GHz to 20 GHz, benefiting from the typical feature of an optoelectronic oscillator structure. The overall phase noise performance of the frequency-tuned output was close to that of the 25-gigahertz fix-frequency RMLL with 320-m loop length, as shown in Figure 7b, where the phase noise curves corresponding to the output of the 10-gigahertz and 20-gigahertz PM-YIG-RMLL and 25-gigahertz SMF-RMLL are compared. The PM-YIG-RMLL showed a slightly better phase noise performance for a frequency offset below 10 kHz, and the 25-gigahertz SMF-RMLL showed better performance above 10 kHz. This may have resulted from the difference in the bandpass filters.

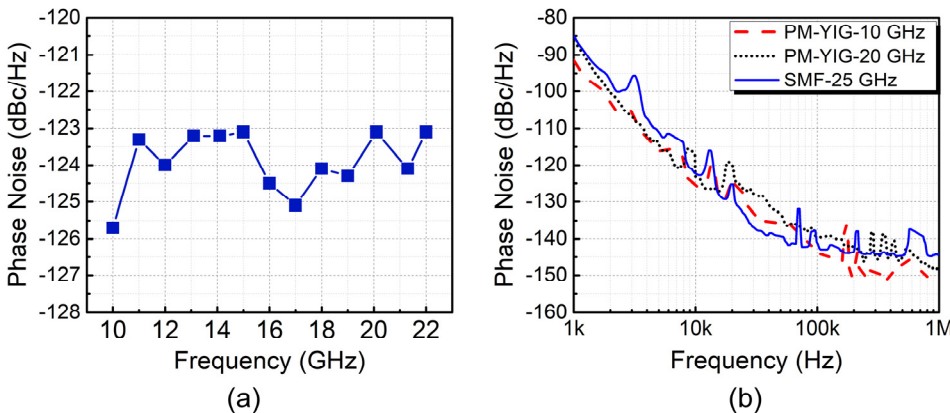

**Figure 7.** The frequency−tunable regenerative mode-locked laser (**a**) phase noise at the 10 kHz offset from the carrier frequency; (**b**) overlapped phase noises of the 10-gigahertz and 20-gigahertz PM−YIG−RMLL and 25-gigahertz SMF−RMLL.

The optical pulse width of the frequency-tunable RMLL was measured to be in the range of 11 ps to 16 ps, as shown in Figure 8a. The calculated timing jitter (integrated from 1 kHz to 1 MHz) for each measured repetition frequency is shown in Figure 8b. The timing jitter for most frequency points was below 10 fs, with the highest timing jitter of 11.7 fs at 10 GHz and the lowest timing jitter of 6.1 fs at 21 GHz.

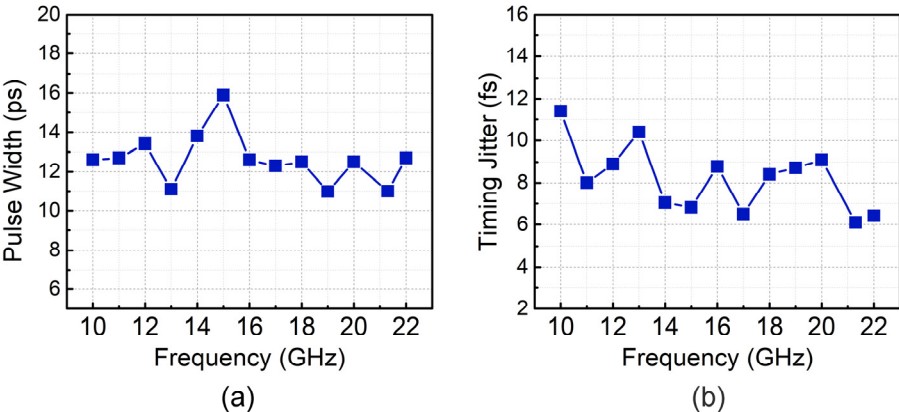

**Figure 8.** The frequency-tunable regenerative mode-locked laser's (**a**) pulse width and (**b**) timing jitter.

## 4. Discussion and Conclusions

A conventional C-band RMLL structure can be constructed by either an EDFA or an SOA. It has been shown that the EDFA-based RMLL has a lower residual-phase noise floor than the SOA-based RMLL [11]. However, the SOA-based RMLL has a better phase noise performance close to the carrier frequency, where the $1/f$ noise dominates. In terms of stability, the EDFA-based RMLL is more susceptible to the influence of the super mode competition [12,14]. Moreover, the SOA is more promising in terms of integrating capacity. Therefore, in the C-band, whether an EDFA or an SOA should be adopted in the RMLL is dependent on specific application requirements. For O-band application, while a praseodymium-doped fiber amplifier (PDFA) can be used to provide gain, it is still not widely available. In addition, the noise figure of the PDFA is 2–3 dB higher than that of best-class EDFA, the phase noise performance of the PDFA-based RMLL needs further investigation. Considering the possible super-mode competition issue with the fiber amplifier, we consider that the SOA is still the most viable option to construct an O-band RMLL. When restricting the gain medium to the SOA, the O-band RMLL reporting in this paper shows similar phase noise performance to that of the C-band RMLL [11].

For the current scheme, the timing jitter of the generated short pulse is limited by the phase noise performance of the YIG filter and the SOA, which have a phase noise level of around −120~−130 dBc/Hz and −130~−140 dBc/Hz, respectively, at 10 kHz from the carrier frequency in most cases. Further optimization can be carried out on the influence of the filter, the amplifier, and the operational condition of the system.

In conclusion, we propose the O-band RMLL, based on an SOA, demonstrated in this paper. The O-band RMLL can generate ultra-low timing-jitter short pulses with no dedicated dispersion control. With a 910-m SMF and a fixed-frequency electrical bandpass filter, 25-gigahertz optical short pulses with pulse width of 7.3 ps and timing jitter of 3.6 fs were realized. With a 300-m PM structure and a current-tuned YIG filter, self-starting-mode locked optical pulses with a repetition rate tunable from 10 GHz to 22 GHz were demonstrated, with a pulse width below 16 ps and a timing jitter in the order of 10 fs. This simple, robust, and high-performance RMLL configuration can potentially be applied in the fields of ADCs, short-reach optical communications, and optical information processing systems.

**Author Contributions:** Conceptualization, D.L. and H.Q.; methodology, H.Q.; formal analysis, H.Q. and Z.Z.; investigation, H.Q.; resources, D.L. and R.Z.; writing—original draft preparation, H.Q.; writing—review and editing, D.L. and L.Z.; visualization, H.Q. and Z.Z.; supervision, D.L. and L.Z.; funding acquisition, D.L. and R.Z. All authors have read and agreed to the published version of the manuscript.

**Funding:** This work was supported by the National Key Research and Development Program of China under Grant No. 2019YFB2203800 and the National Natural Science Foundation of China (NSFC) under Grant No. 62074141.

**Institutional Review Board Statement:** Not applicable.

**Informed Consent Statement:** Not applicable.

**Data Availability Statement:** Not applicable.

**Conflicts of Interest:** The authors declare no conflict of interest.

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
