# Peer review of "O-Band Frequency-Tunable (10–22 GHz) Ultra-Low Timing-Jitter (<12-fs) Regenerative Mode-Locked Laser"

_photonics, doi:10.3390/photonics9030169_

Round 1

Reviewer 1 Report

The authors demonstrate an O-band regenerative mode-locked laser (RMLL) based on an optoelectronic oscillation configuration and SOA. With single-mode fiber, 25-GHz optical pulses were realized. With a 300-m polarization-maintaining fiber and a tunable filter, low-timing jitter mode-locked optical pulses with repetition-rate tunable from 10 GHz to 22 GHz were achieved. This work is interesting, and this article is well organized. It can be accepted. Several suggestions:

  • Please discuss the difference (e.g., the performance, the noise source) between the O-band RMLL and the C-band RMLL.
  • Please briefly discuss the limitation of the proposed scheme. For example, how to obtain pulses with lower timing-jitter? What is the limitation of the timing-jitter for this scheme?

Reviewer 2 Report

This manuscript domonstrated a frequency-tunable and low-timing-jitter O-band regenerative mode-locked laser using an optoelectronic oscillation configuration and electric controlled Yttrium Iron Garnet (YIG) bandpass filter. It is interesting and well presented. Just two issues need to be clarified.

  1. The center wavelength of the 25GHz RMLL was around in 1337 nm, while the center wavelength of the10GHz was set at about 1333.5 167 nm. What is the reason?
  2. An optical filter was incorporated in the RMLL, is it possible to tune freely the central wavelength of the RMLL?
